# A New Method to Evaluate Lower Esophageal Distension Capacity in Eosinophilic Esophagitis by Using Functional Lumen Imaging Probe (EndoFLIP™)

**DOI:** 10.3390/diagnostics14020218

**Published:** 2024-01-19

**Authors:** Sergio Casabona-Francés, Ancor Sanz-García, Guillermo J. Ortega, Cecilio Santander, Teresa Perez-Fernandez, Pedro Majano, José Maria Olalla, Rocio Juárez-Tosina, Ralph Mueller, Stephen Attwood, Alfredo Lucendo

**Affiliations:** 1Department of Gastroenterology and Hepatology, Hospital Universitario de La Princesa, 28006 Madrid, Spain; scasabonafrances@gmail.com (S.C.-F.); cecilio.santander@salud.madrid.org (C.S.); tperfer@gmail.com (T.P.-F.); pmajano@gmail.com (P.M.); 2Biomedical Research Networking Center in Hepatic and Digestive Diseases (CIBERehd), 28029 Madrid, Spain; ajlucendo@hotmail.com; 3Instituto de Investigación Sanitaria Princesa (IIS-IP), 28006 Madrid, Spain; 4Data Analysis Unit, Instituto de Investigación, Sanitaria Hospital Universitario de la Princesa, 28006 Madrid, Spain; ancor.sanz@gmail.com; 5Consejo Nacional de Investigaciones Científicas y Técnicas, CONICET, Buenos Aires C1425FQB, Argentina; 6Science and Technology Department, National University of Quilmes, Bernal B1876BXD, Argentina; 7Department of Pathology, Hospital General La Mancha Centro, 13600 Alcázar de San Juan, Spain; jomaolalla@yahoo.es (J.M.O.); roci_med@hotmail.com (R.J.-T.); 8Department of Clinical Research and Development, Dr. Falk Pharma GmbH, 79108 Freiburg, Germany; ralph.mueller@drfalkpharma.de; 9Department of Health Services Research, Durham University, Durham DH1 3LE, UK; seaattwood@gmail.com; 10Department of Gastroenterology, Hospital General de Tomelloso, 13700 Tomelloso, Spain; 11Instituto de Investigación Sanitaria de Castilla-La Mancha (IDISCAM), 45071 Toledo, Spain

**Keywords:** functional lumen imaging probe, impedance planimetry, Eosinophilic Esophagitis, esophageal distensibility, dysphagia, EndoFLIP^TM^, esophagus

## Abstract

Endoluminal functional lumen impedance planimetry (EndoFLIP^TM^) has become the gold standard to evaluate esophageal distensibility, although the study itself and its analysis present challenges. We propose here a new method to assess lower esophageal distension capacity that overcomes several limitations of prior approaches, including incomplete and corrupted EndoFLIP^TM^ recordings. Esophageal distension capacity was evaluated with a 16-channel EndoFLIP^TM^ in 10 controls and 14 patients with eosinophilic esophagitis (EoE). Controls were evaluated once. EoE patients were evaluated at baseline and after at least six weeks of treatment with orodispersible budesonide tablets, 1 mg bd. Balloon volumes were increased by 5 mL stepwise, either reaching a maximum volume of 60 mL or a maximum balloon pressure of 60 mmHg. Recordings were analyzed with a homemade R script. The mean esophageal diameter at 60 mL, D (60 mL), was calculated or extrapolated depending on whether the 60 mL volume was reached. By fitting a Michaelis–Menten curve across all measured diameters throughout all constant volume steps, the mean D (60 mL) was estimated. For control subjects, the mean ± SD value of D (60 mL) was 17.08 ± 1.69 mm, and for EoE patients at baseline, D (60 mL) was 14.51 ± 2.68 mm. After six weeks of treatment of EoE patients, D (60 mL) significantly increased to 16.22 ± 1.86 mm (paired Wilcoxon signed test: *p* = 0.0052), although the values for control subjects were not reached. The estimated mean esophageal diameter at 60 mL is a good proxy for esophageal distension capacity, which correlates with clinical outcomes in EoE. The method presented in this study overcomes difficulties encountered during the standard measurement protocol, allowing the analysis of recordings from incomplete and corrupted registries.

## 1. Introduction

The techniques used to investigate gastroesophageal sphincter physiology have evolved from pull-through and stationary manometry to the more recent high-resolution manometry [1]. The functional luminal imaging probe EndoFLIP^TM^ (Medtronic, Minneapolis, MN, USA) has taken center stage in recent years due to its ability to overcome the poor correlation between sphincter strength and competence evidenced by manometry and because of its clinical utility as a diagnostic tool to guide and measure response to therapy [2]. It consists of an impedance planimetry system that registers data from a functional luminal image probe included in a catheter contained within a balloon in which a conductive fluid is infused. An integrated pressure sensor measures intrabag pressure [3].

Since its introduction, impedance planimetry measuring multiple adjacent cross-sectional areas through EndoFLIP^TM^ [4] has allowed the evaluation of several esophageal pathological conditions as well as their evolution [5,6,7]. Despite the fact that EndoFLIP^TM^ was originally developed to investigate the esophago-gastric junction (EGJ), it is also used in other sphincteric and non-sphincteric regions of the gastrointestinal tract [2]. EndoFLIP^TM^ allows the assessment of intraluminal esophageal geometry and mechanics as a safe and easy-to-apply method while endoscopy is being performed by providing a three-dimensional image of the esophageal lumen using high-resolution impedance planimetry to measure changes in pressure, diameter, and volume.

Eosinophilic esophagitis (EoE) is a chronic inflammatory disease that causes esophageal dysfunction and infiltration of the esophagus by eosinophils [8]. The natural course of EoE is chronic and appears to be progressive, with long-standing eosinophilic inflammation leading to esophageal remodeling with stricture formation and functional damage in the long term [9,10]. Despite its potential utility in the evaluation of patients with EoE, the EndoFLIP^TM^ is still being evaluated in this condition and is not yet included in clinical practice. The place of EndoFLIP^TM^ in severity assessment and therapeutic monitoring as well as its possible superiority to esophageal biopsies to guide EoE treatment remain unclear [11]. After being introduced by oral or nasal route, the EndoFLIP^TM^ catheter is positioned within the esophagus. As the balloon inflates and the esophagus is distended, it also shortens, thus varying the position of the catheter through the lower sphincter. In addition, the esophageal contraction induced when the esophagus is distended by the FLIP makes it difficult to calculate distension capacity, as it often moves during the study.

The current EndoFLIP^TM^ analysis method proposed by Pandolfino et al. [12,13] has several drawbacks that may not only hinder its effectiveness but also prevent the numerical analysis of the study recording. These include (1) the requirement for a strict protocol of increasing volumes at fixed steps until the maximum diameter at 60 mL distension is achieved, which cannot always be accomplished in patients with severe strictures and low-caliber esophagus; (2) the exclusion of data from movements of the catheter with respect to the esophageal wall from the analysis; (3) the lack of consideration for deviations of a monotonic pressure-volume relationship; and (4) missing constant volume regions. Improving the clinical applicability of the technique requires the definition of a new utilization protocol based on the disease of interest and the development of new software to process the data obtained after the study in order to produce more reliable and reproducible results. 

In this study, we present a new method to evaluate esophageal distension capacity through the analysis of EndoFLIP^TM^ recordings that makes the use of the standard and de facto established protocol of increasing volumes stepwise unnecessary. Moreover, the approach presented here deals with several issues not addressed in the previously described analysis method.

## 2. Materials and Methods

### 2.1. Subjects

Two groups of subjects were included in this study. The first group was composed of 10 healthy asymptomatic (i.e., free of esophageal symptoms, including dysphagia, heartburn, and chest pain) adult (>17 years) men (60%) and non-pregnant women volunteers without a history of surgery. The mean age was 31.9 years (range 24–61 years) and the mean body mass index (BMI) was 23.08 kg/cm^2^ (range 17.51–27.07 kg/cm^2^); one had rhinitis, none had asthma, none had dermatitis, one had food allergies, and none had a family history of EoE. They attended the Endoscopy Unit of the Digestive System Service of the Hospital Universitario de la Princesa (Madrid) and participated as a control group; they were indicated for an oral panendoscopy for reasons unrelated to esophageal pathologies and did not receive any compensation for their participation in the study. All of them had a normal endoscopic examination of the esophageal mucosa and multiple esophageal biopsies that excluded microscopic inflammation, eosinophilic infiltration, and additional histopathological alterations. No efforts were made to exclude food allergies or atopy in the control group.

The second group was composed of 14 consecutive patients suffering from active EoE who participated in a trial [14] and received active treatment with orodispersible budesonide tablets at a dose of 1 mg twice daily for 6 weeks. EoE diagnoses were based on the criteria proposed by Liacouras et al. [15]: symptoms of esophageal dysfunction and infiltration of ≥15 eosinophils in at least 1 high-magnification field (400×) derived by at least 6 esophageal biopsies and no clinico-histological response to proton pump inhibitor therapy. Symptoms at baseline and after therapy were measured with the aid of the un-validated Straumann Dysphagia Index (SDI) [16] and the validated EoE Activity Index—Patient-Reported Outcome (EEsAI-PRO) score [17]. Endoscopic features were evaluated according to the EREFS scoring system, which graduates the most common endoscopic findings that characterize EoE: edema, rings, exudates, furrows, and stricture [18]. EndoFLIP^TM^ assessment was performed at baseline and during endoscopic exams performed at the end of treatment.

The study protocol was approved by the ethics committee for research at the Hospital Universitario de La Princesa (act number 22/2017). The study was conducted according to the guidelines of the Declaration of Helsinki. All patients and control subjects were adults (>18 years) and provided written informed consent before any study procedures.

### 2.2. EndoFLIP^TM^ Measurement Protocol

All subjects underwent upper endoscopy in the left lateral decubitus position after a minimum of 6 h of fasting. Propofol-based sedation was provided during the procedure. The EndoFLIP^TM^ distension capacity measurements were performed during the examination of esophageal appearance and before esophageal biopsies were collected. The FLIP (EndoFLIP^TM^ EF-322N; Medtronic, Minneapolis, MN, USA) was 16 cm in length with a 3 mm diameter catheter (9F). A catheter containing a distensible bag covering the distal part with 16 pairs of metal impedance rings 10 mm apart and a solid-state pressure transducer was purged and calibrated to atmospheric pressure before transoral probe placement, according to the manufacturer’s instructions. Each procedure was identified with a unique code with no identifiable patient details. The endoscopist performing exams was the only researcher able to identify which patient each study came from. Alarms for inflation volume, inflation rate, and pressure were all set to 60 mmHg. The EndoFLIP^TM^ catheter was positioned into the esophageal body parallel to the endoscope and slid alongside the scope until 1 to 3 impedance sensors were observed below the esophagogastric junction (EGJ). The endoscope was removed during the EndoFLIP^TM^ measurement to avoid interference with the balloon.

Throughout the whole paper, we use the term “distension capacity” instead of the most commonly—although incorrectly—employed “distensibility”, which formally indicates a relative change in size due to changes in pressure, a measure which is hard to evaluate indeed. Because the method proposed here only evaluates the mean diameter at an actual (reached) or theoretical volume of 60 mL, we decided to employ the term “distension capacity” instead of “distensibility” because it seemed more appropriate.

The balloon was inflated to 30 mL, and the position of the catheter was adjusted until the lower esophageal sphincter (LES) was over the 2 most distal electrodes, making sure that the LES was visible near the bottom of the EndoFLIP^TM^ image. The catheter was held tightly to prevent the balloon from slipping into the stomach and was controlled to position the balloon so that the LES was visible near the bottom of the EndoFLIP^TM^ image on the screen.

The balloon was then deflated to 20 mL, and 5 mL stepwise distension starting from 20 mL was performed to the target volume of 60 mL. Each stepwise distension volume was maintained for at least 20 s. The balloon distension protocol ended either when the pressure read over 60 mmHg or the balloon volume was 60 mL, whichever came first. The minimum diameter in the esophageal body was recorded at increasing volumes (20 to 60 mL) of the infusion solution and pressure (mmHg) in the distensible balloon during periods of no swallowing movements and no esophageal body contraction. After completing the stepwise inflation, data were saved for subsequent analysis, and the balloon was then deflated and removed from the patient.

### 2.3. Data Analysis: Prepossessing Steps

For the analysis, EndoFLIP^TM^ data were exported to ASCII files, including the time series of pressure, volume, and the measured esophageal diameters sampled at 10 Hz; examination ID code, record times, pump status, channel numbers from #1 to #16, balloon pressure (BP), and balloon volume (BV) were the variables considered. These files were read by homemade software in R [19] and pre-processed accordingly. First, the part of interest in each examination was selected from the whole record by looking at the volume levels; that is, the complete “stairway” of steady volumes (20 to 60 mL or the maximum volume reached) was visually selected, and the corresponding dataset was stored for further analysis.

A prepossessing routine of the time series analysis encompassed the detection and replacement of outliers. The number of outliers was always well below 5% of the total records. A running smoothing filter averaging over temporal windows of 20 points (two seconds) was applied.

The identification and selection of the steady-volume regions of approximately 20 s (200 data points) were performed. When these regions were not ordered in ascending order of volume, they were reordered, along with their corresponding pressures and diameters, to obtain the arrangement that was used for the final analysis. These steady-volume regions are the most stable and allow for a better assessment of the esophageal distension capacity by excluding pressure variations due to external factors by minimizing dynamic movements produced by balloon inflation and deflation. These regions, with their corresponding pressure and diameter data, were exclusively used for distension capacity assessments.

A typical representation of the outcome of an EndoFLIP^TM^ measurement is plotted in Figure 1. The upper panel of this figure represents, as a topographic map, the measured diameters of each channel, with dark colors representing smaller diameters and blue colors representing larger diameters. The lower panel of Figure 1 displays the procedure times for both the balloon volume and pressure.

### 2.4. Data Analysis: Caveats

Several factors make the automated analysis of recordings difficult; these include:The movement or “sliding” of the EndoFLIP^TM^ catheter with respect to the esophageal wall; this is mainly caused by increasing pressure within the balloon, which pushes the esophagus upward. When this situation occurs, the impedance planimetry diameters change their original positions, resulting in a new position located a certain distance from the original one. Figure 2 illustrates this situation.In some circumstances, the EndoFLIP^TM^ balloon may be twisted or squeezed, and the pressure sensor located at the tip of the balloon records a pressure that is not the uniform pressure along the whole balloon. Therefore, it is expected that, regardless of the presence of peristaltic or anti-peristaltic contractions, under these circumstances, a theoretical monotonic volume–pressure relationship will not be achieved. Figure 3 shows two examples of “good” (upper panel) and “poor” (lower panel) volume–pressure relationships.Missing constant-volume steps. In some situations, some constant volume steps cannot be obtained due to technical, operational, and/or physiological factors; therefore, a clean “stair” cannot be constructed to evaluate the esophageal distension capacity.Lastly, due to a lack of an appropriate esophageal caliber in disease states, high volumes, i.e., greater than 45 mL, are not achievable in some cases, thus impeding the achievement of a diameter estimate at the target 60 mL volume.

### 2.5. Data Analysis: Assessment of Distension Capacity

All these caveats were prevented and resolved in the following way:The sliding of the EndoFLIP^TM^ with respect to the esophageal wall was corrected to ensure that the EndoFLIP^TM^ sensors were always measuring the same esophageal wall diameter throughout the study. To achieve this, we first localized, among the set of steady-volume areas, the LES at its highest position during the study, which was typically at the study’s end. In the case of Figure 2, this was located around channel #8, as marked with the white circle number 4; this was called the “reference region”. After that, we selected the channels located at least 2 cm above where the LES was located (channel #8), channels #1–#6, which comprised the region selected to study. Thereafter, we localize the LES position in all the constant volume regions. Figure 2 shows some of them marked with circles numbered 1, 2, and 3. During the whole study, the LES moved from channel #13 at the beginning to channel #8 at the end. This means that all the planimetry sensors shifted from their original position upward by 5 cm; that is, the LES was measured with channel #13 at the start, but it was measured with channel #8 at the end of the study. Throughout the study, the channel measuring the LES and the remaining ones were shifted upward in the following way: from step 1 to step 2 in 1 channel (13 to 12), from step 2 to step 3 in 2 channels (12 to 10), and from step 3 to step 4 in 2 channels (10 to 8). This means that fixed areas of the esophageal wall were measured by different sensors throughout the study, and therefore, a transformation of the sensors needed to be performed. This is readily seen by the shadowed rectangles in Figure 2, which shows the 6 channels that measured a delimited area change throughout the study.Only those regions in which an appropriate volume–pressure relationship was obtained were evaluated, discarding those in which a monotonic relationship was not documented. The Kendall τ correlation provides a good estimate of the monotonic volume–pressure relationship.As the method of evaluation of the theoretical maximum diameter does not need all the constant volume steps, we only used data related to appropriate constant volume steps with a minimum number of required steps of three.Also, in the case of the impossibility of reaching higher constant volumes due to dangerous high pressures (greater than 60 mmHg), for instance, we estimated the theoretical diameter at a 60 mL volume.

Considering all these issues, we developed a method to evaluate the esophageal distension capacity based on the estimation of the theoretical diameter reached at a maximum volume of 60 mL. To achieve this, we selected the regions of constant volume accordingly with the selection processes outlined in (2) and the channels accordingly with the selection process outlined in (1). With this information, we fitted a Michaelis–Menten curve, D = (V_0 V)/(K + V), where V is the balloon volume, D is the diameter, V0 is a constant related to the maximum possible diameter reached (plateau), and K is a constant. K should be estimated after several runs. In our case, we fixed V0 = 20 and K = 25. Figure 4 shows two typical examples of the regression of the diameter as a function of volume for the case of lacking constant volume levels (left panel) and a complete case in the right panel.

To analyze the changes in a certain diameter of the esophageal wall, the displacement of the catheter throughout the procedure must be considered since a certain esophageal diameter is measured by several sensors, as in the case explained above with the LES. The method for this is by “correcting” this shift in the balloon sensors step by step. The esophagus wall diameters measured by channels #3 to #6 in region 4 (the white circle in Figure 2), for instance, were then measured by channels #4 to #7 in region 3 (white circle, Figure 2) because a shift of approximately 1 channel existed between these two regions. A visual description of the method is depicted in Appendix A, with the accompanying description in Appendix A.

### 2.6. Statistical Analysis

Means ± SDs or medians ± interquartile ranges (IQRs) were used for continuous variables, and percentages were used for categorical variables. Intra-subject comparisons were made using the paired *t*-test or Wilcoxon rank-sum test; groups were compared using the *t*-test or Mann–Whitney U test for continuous variables and the χ^2^ test or Fisher exact test for categorical variables. All analyses were carried out using SPPS v.18.0 (SPSS Inc. Armonk, NY, USA) and the R language [19]. A *p*-value < 0.05 was considered significant.

## 3. Results

### 3.1. Subjects

The demographic, clinical, endoscopic, and histological characteristics of patients and controls are summarized in Table 1.

Control subjects (6 men) had a median (IQR) age of 27 (24–61) years and a body mass index (BMI) of 24.02 (17.51–27.07) kg/m^2^. Only one subject suffered from rhinitis and a food allergy. None of the controls had dysphagia, vomiting, or weight loss. No control subjects presented symptoms of esophageal features of gastroesophageal reflux disease (GERD) or EoE at endoscopy, and esophageal biopsies presented no eosinophilic infiltration or additional pathological changes.

EoE patients (11 males) had a median (IQR) age of 19 (13–43) years and a BMI of 22.68 (16.87–30.04) kg/m^2^. Seven EoE patients presented a fibro-stricturing phenotype, and another seven presented an inflammatory one, as defined by endoscopic features [18]. No EoE patients had endoscopic signs of GERD. Regarding comorbidities, 11 out of the 14 presented concomitantly rhinitis, 8 presented with asthma, 3 presented with dermatitis, and 5 presented with food allergies. A family history of EoE was present in three patients. As for symptoms at baseline, 10 patients reported dysphagia, 10 reported esophageal impactions in the past, 3 reported vomiting, and 2 reported weight loss. Nine patients had previously taken corticosteroid treatment with inhaled metered-dose fluticasone propionate, and two of them had also required endoscopic dilation in the past. 

The clinical, endoscopic, and histological data in patients with EoE as well as changes induced by therapy are summarized in Table 2.

Treatment with orodispersible budesonide tablets significantly reduced peak eosinophil counts in both the proximal and distal esophagus and significantly improved symptoms according to measures provided by the SDI and the EEsAI score. Endoscopic features were also reversed. 

There were no serious complications associated with the use of the EndoFLIP^TM^ device in any of the patients or controls. Mild self-limited discomfort (retrosternal pain, odynophagia, heartburn, and low-grade fever) was reported by some patients. They could not be related to the performance of this technique and/or the endoscopy but represent a bias when specifically asking for retrosternal discomfort after the procedure.

### 3.2. Esophageal Distension Capacity Index

The mean ± standard deviation esophageal diameter at 60 mL balloon distension (D (60 mL)), whether calculated or extrapolated, is presented in Appendix A for control subjects, in Appendix A for EoE patients with active disease, and Appendix A for EoE patients after treatment. Along with the diameters, additional variables obtained during the study were also considered, consisting of the maximum pressure reached and the maximum balloon volume achieved. For control subjects, the mean ± SD values of the maximum pressure reached in the controls were 46.99 ± 9.78 mmHg; as for active EoE patients, they were 64.35 ± 14.11 mmHg (*p* = 0.019 vs. controls), which decreased to 57.40 ± 14.55 mmHg after effective treatment (*p* = 0.267 vs. before treatment). These figures show that the study could be completed at lower pressures in control subjects. After inducing remission, EoE patients exhibited lower pressures than when they had active disease, although the difference did not achieve statistical significance (*p* = 0.267).

The maximum intra-balloon volume achieved during the study showed comparable outcomes: in control subjects, the mean ± SD balloon volume was 57.50 ± 4.24 mL. In EoE patients with active disease, the mean ± SD volume was 55.71 ± 8.05 mL before treatment and increased to 58.57 ± 4.12 mL after treatment. Again, this increase in volumes induced by anti-inflammatory therapy did not achieve significance (*p* = 0.134).

Both parameters—the maximum pressure reached and maximum volume achieved—indicate a decreased esophageal distension capacity in active EoE patients that cannot be used as trustable parameters to identify this property. Thus, we turned our attention to the esophageal diameter at the fitted/actual maximum volume of 60 mL, abbreviated as D (60 mL). For control subjects, the mean ± SD D (60 mL) was 17.08 ± 1.69 mm. Patients with active EoE had a D (60 mL) of 14.51 ± 2.68 mm, which significantly increased after treatment up to 16.22 ± 1.86 mm (*p* = 0.0053).

Figure 5 presents the differences between the three study groups.

### 3.3. Contractile Pattern

The presence of esophageal contractions, whether antegrade or retrograde, produced when the esophagus is distended by the FLIP is an issue of concern when EndoFLIP^TM^ recordings are analyzed. Absent contractility was registered in 5/12 EoE patients before treatment and in none of the patients with EoE in remission after treatment with budesonide. Retrograde esophageal contractions were not observed.

### 3.4. Relationship between D60 and EEsAI

The question of whether a relationship between esophageal distension capacity, as evaluated by D (60 mL), and the EEsAI index is of interest at the time of predicting the clinical changes after treatment. To assess the existence of any relationships between both tests, the correlation between the changes between both variables—D (60 mL) and the EEsAI-PRO score—before and after the treatment was calculated in the ten patients who had both variables measured, and it was −0.464 (t = −1.8176, df = 12, *p* = 0.094). Although not significant at 5%, this calculation shows an approximately inverse linear relationship between both variables.

## 4. Discussion

This research describes a new numerical procedure to analyze EndoFLIP^TM^ recordings specifically developed to minimize most of the difficulties presented during a typical study. Although the EndoFLIP^TM^ system provides rich information on the esophageal contractibility and function induced by its distension, the analysis and interpretation of the data it produces should be carried out with caution. Certainly, there exists a gap between the theoretical and practical results obtained using this device. In this work, we highlighted at least four situations in which a deviation from the theoretical use of the EndoFLIP^TM^ should be considered and corrected. The first one is related to the movements of the EndoFLIP^TM^ balloon with respect to the esophagus wall; the second one is the limitation to reaching high balloon volumes, e.g., 60 mL, which occurs in some, especially pathological, cases; the third is that a deviation from an increasing monotonic volume–pressure relationship exists, in particular during the firsts fixed-volume stages; and the last is that some fixed-volume steps may be skipped due to technical reasons, generating an incomplete fixed-volume staircase. These issues are usually present, one or several at the same time, in typical clinical examinations, and they are a source of contamination of the results or even obstruct the numerical analysis of the whole study. This fact is most obvious when a strict protocol of increasing the volume 5 mL stepwise is followed when trying to achieve a maximum volume of 60 mL, as has been carried out in most studies published.

Patients suffering from EoE were the subjects with the narrowest D (60 mL) caliber (14.51 ± 2.68 mm), but this value improved after six weeks of effective treatment with orodispersible budesonide tablets, achieving a D (60 mL) value of 16.22 ± 1.86 mm. On the other hand, control subjects displayed the maximum D (60 mL) value (17.08 ± 1.69 mm).

Although all studies in controls and EoE patients were originally scheduled to be performed under the 5 mL increasing volumes protocol, in many of them, as can be observed in Appendix A, it was not possible to complete this goal in most cases due to the high pressures attained even at lower volumes, such as 35 mL (as in patient P3BT in Appendix A). The method we developed allowed us to deal with these cases. In fact, it is possible to estimate the mean diameter at 60 mL distension (D (60 mL)) with at least three constant volume steps, as happened with the aforementioned patient. The fact that in most patients analyzed the limiting volume of 60 mL was reached does not mean that all the fixed-volume steps were used; in many of these cases, the monotonic volume–pressure relationship was not followed in some parts of the measurement; these parts were discarded for the numerical analysis, and only some of the constant volume regions remained. We even propose that modifying the balloon inflation steps to 10 mL intervals could simplify the recording protocol and reduce the examination length (and sedation) while yielding similar results.

Another issue that should be mentioned is the presence of contraction waves, anterograde or retrograde, during the study. These waves last approximately 3–5 s and produced large variations in the esophagus diameter and, consequently, in the EndoFLIP^TM^ records. Taking into account that each constant volume was 20 s, this means that at least four waves were “averaged” in each channel during the study, giving a mean value of the maximum and minimum dilatation that, to our consideration, is comparable with the mean value of the esophagus in the absence of contractions.

The EndoFLIP^TM^ is proposed as a potentially useful tool in the evaluation of EoE. Eosinophil infiltrate density has been shown not to predict outcomes in patients with EoE. By contrast, esophageal distension capacity, which does not correlate with eosinophil density [13], may define fibrostenotic severity and clinical outcomes [20]. Patients with reduced distension capacity are at risk of food impactions, as reduced esophageal compliance (<225 mm^2^) was shown to be a good predictor of esophageal food bolus impaction and the need for esophageal dilatation. In this study, we showed that the change in D (60 mL) is a good parameter to predict clinical response, which was proven by its parallel changes with esophageal symptoms (measured with the EEsAI-PRO instrument) and independence of histological activity measured by active eosinophilic inflammation in esophageal biopsies. This opens the possibility of performing an evaluation of response to treatment by assessing distension capacity instead of the classical histological response.

This study has several limitations, however. The first one is that the method was specifically developed to assess distension capacity in the lower part of the esophagus, as the LES was used as a landmark; in this sense, its potential to evaluate other parts of the esophagus is unknown. Also, the method could be improved by weighting in some way the robustness of the fitted curve according to the number of constant volumes employed in the fitting, an issue we are working on. Lastly, a more accurate estimation of esophageal distension capacity should be provided by considering not only the diameter at 60 mL (D (60 mL)) with the 16 cm catheter (EndoFLIP^TM^ EF-322N) but also a corresponding mean pressure at that volume. We think that both parameters would give a more robust idea of the actual esophageal distension capacity.

## 5. Conclusions

This research is a proposal to facilitate the method of analysis of the data recorded during the study of esophageal distension capacity, ensuring the homogeneity of the studies, the comparison of the results, and, in short, increasing the dissemination of the EndoFLIP^TM^ to improve the monitoring and follow-up of patients with EoE.

We are aware that much more data can now be collected and analyzed by EndoFLIP^TM^, including the analysis of esophageal body contractions in response to balloon inflation and even the opening of the GEJ. New efforts are required to standardize the method and promote further developments.

## Figures and Tables

**Figure 1 diagnostics-14-00218-f001:**
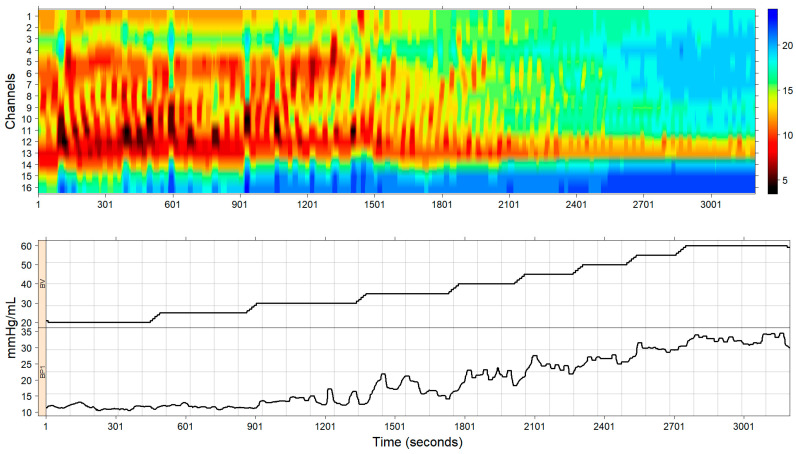
Typical representation of an EndoFLIP^TM^ measurement. The upper panel shows a topographic representation of the impedance planimetry diameters color-coded accordingly with the right bar color key. This representation shows measured diameters in each of the 16 channels. Also, both antegrade, around time = 301 s, and retrograde, around time = 1801 s, contractions were observed. The lower panel shows the evolution of the balloon volume and pressure throughout the study.

**Figure 2 diagnostics-14-00218-f002:**
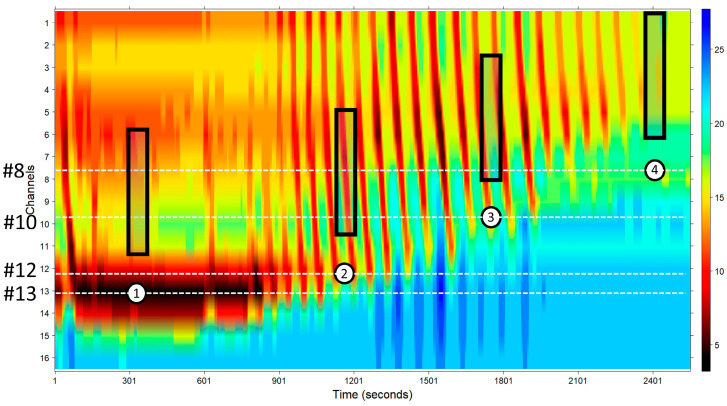
EndoFLIP^TM^ movements with respect to the esophagus wall. This figure shows how the EndoFLIP^TM^ balloon slides with respect to the esophagus wall. White circles show the EGJ location in several instances in the study. Dotted white lines mark approximately which channels recorded the corresponding diameters. Shadowed black rectangles represent the approximate esophagus area recorded by different sensors.

**Figure 3 diagnostics-14-00218-f003:**
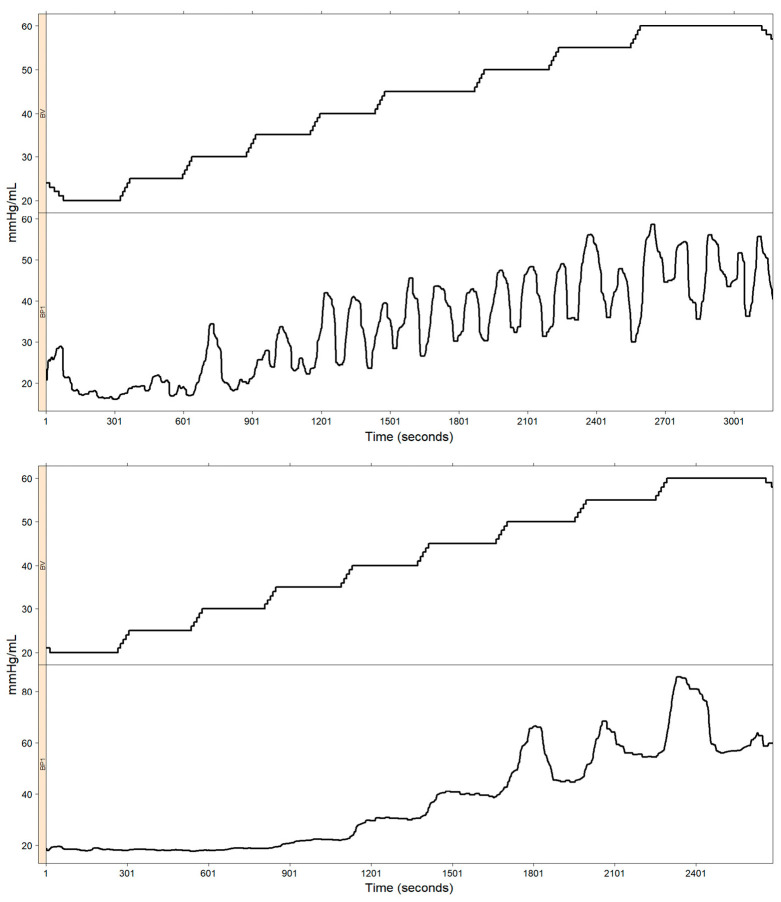
Volume–pressure relationship. The upper panel shows an approximately monotonic relationship between the volume balloon and the intra-balloon pressure throughout the study. The lower panel shows a “failed” monotonic relationship between volume and pressure. Only in the last part of the study, from time = 1201 s on, there existed an approximately monotonic relationship between volume and pressure.

**Figure 4 diagnostics-14-00218-f004:**
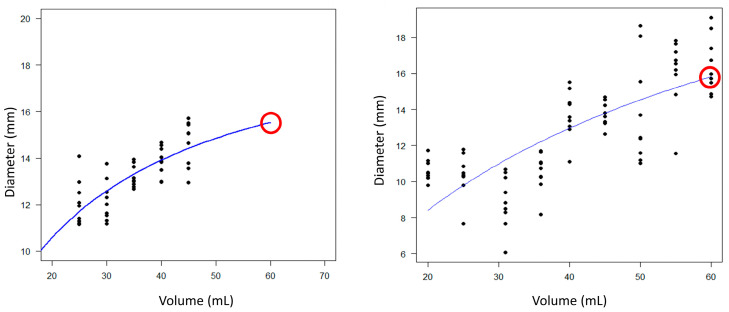
Regression of a theoretical Michaelis–Menten curve (blue solid line) across of all the impedance planimetry diameters for the selected constant-volume areas and channels. Red circles show the predicted diameter at the maximum volume of 60 mL. Left panel shows the case of lacking higher volumes. The right panel shows the case of a complete set of constant-volume sections.

**Figure 5 diagnostics-14-00218-f005:**
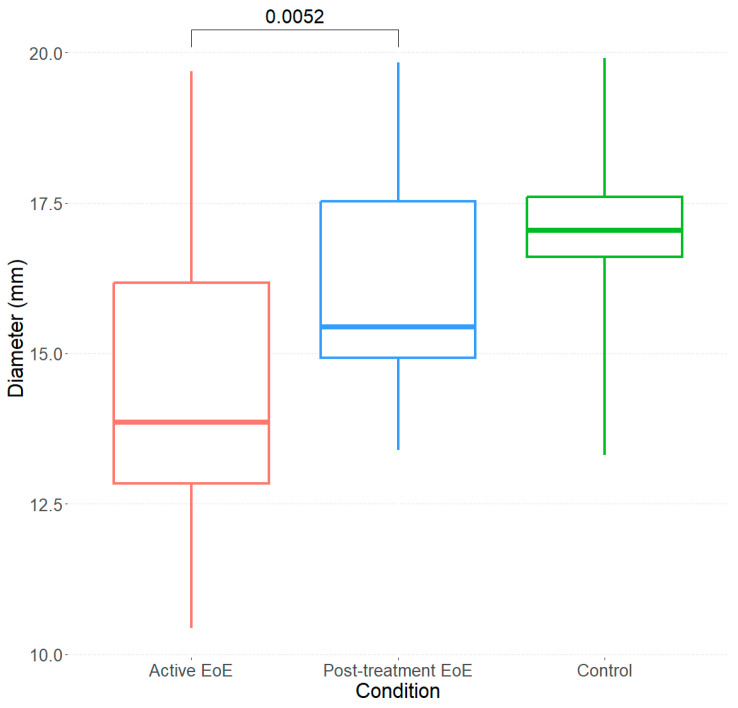
Boxplots of D (60 mL) for the three analyzed cases. The red boxplot corresponds to the 14 EoE patients at baseline with a mean diameter of 14.51 ± 2.68 mm. The blue boxplot corresponds to the 14 patients after the treatment with a mean diameter of 16.22 ± 1.86 mm. The green boxplot corresponds to the cases of ten control subjects with a mean diameter of 17.08 ± 1.69 mm.

**Table 1 diagnostics-14-00218-t001:** Demographic, clinical, endoscopic, and histological data in healthy controls and patients with EoE.

	Control (*n* = 10)	EoE Patients at Baseline (*n* = 14)
Age (years); median (IQR)	27 (24–61)	19 (13–43)
Male; *n* (%)	6 (60)	11 (79)
BMI (kg/m^2^); median (IQR)	24.02 (17.51–27.07)	22.68 (16.87–30.04)
Rhinitis; *n* (%)	1 (10)	11 (79)
Asthma; *n* (%)	0 (0)	8 (57)
Dermatitis; *n* (%)	0 (0)	3 (21)
IgE-mediated food allergy; *n* (%)	1 (10)	5 (36)
Family background of EoE; *n* (%)	0 (0)	3 (21)
Food impaction; *n* (%)	0 (0)	10 (71)
Dysphagia; *n* (%)	0 (0)	10 (71)
Vomit; *n* (%)	0 (0)	3 (21)
Weight loss; *n* (%)	0 (0)	2 (14)
Prior corticosteroid Tx; *n* (%)	0 (0)	9 (64)
Prior esophageal dilation; *n* (%)	0 (0)	2 (14)
Straumann Dysphagia Index (range: 0 to 9 points); median (IQR)	1 (1–1)	9 (3–13)
EREFS score (range: 0 to 9 points); median (IQR)	0 (0–0)	4 (2–8)
Peak eosinophil count at proximal esophagus, cells/high power field; median (IQR)	0 (0–0)	40 (17–120)
Peak eosinophil count at distal esophagus, cells/high power field; median (IQR)	0 (0–0)	53 (28–100)
Discomfort after EndoFLIP^TM^, *n*	7	1

IQR, interquartile range; BMI, body mass index; EoE, eosinophilic esophagitis, EREFS, endoscopic reference score, EndoFLIP^TM^, endoscopic functional lumen imaging probe.

**Table 2 diagnostics-14-00218-t002:** Clinical, endoscopic, and histological data in patients with EoE before and after a 6-week treatment with orodispersible budesonide tablets.

	EoE Patients at Baseline (*n* = 14)	EoE Patients Post Treatment (*n* = 14)	*p*
Straumann Dysphagia Index; median (IQR)	9 (3–13)	1 (1–6)	<0.001
Esophageal dilation; *n*	2	0	0.241
EEsAI-PRO score; mean ± SD (range)	49 ± 28 (15–80)	12 ± 25 (0–42)	<0.001
EREFS score; median (IQR)	4 (2–8)	0 (0–4)	<0.001
Edema; *n*			<0.001
0: Distinct vascularity	1	13	
1: Decreased	13	1	
2: Absent	0	0	
Rings; *n*			0.384
0: None	9	10	
1: Mild	1	3	
2: Moderate	3	1	
3: Severe	1	0	
Exudates; *n*			<0.001
0: None	1	13	
1: Mild	3	1	
2: Severe	10	0	
Furrows; *n*			<0.001
0: None	0	13	
1: Mild	14	1	
2: Severe	0	0	
Strictures; *n*			0.049
0: Absent	10	14	
1: Present	4	0	
Crêpe paper; *n*			0.241
0: Absent	12	14	
1: Present	2	0	
Peak eosinophil count at proximal esophagus, cells/high power field; median (IQR)	40 (17–120)	0 (0–5)	<0.001
Peak eosinophilic count at distal esophagus, cells/high power field; median (IQR)	53 (28–100)	0 (0–16)	<0.001
Discomfort after EndoFLIP^TM^; *n*	1	0	0.5

IQR, interquartile range; EesAI-PRO, eosinophilic esophagitis activity index—patient-reported outcome; EoE, eosinophilic esophagitis, EREFS, endoscopic reference score, EndoFLIP^TM^, endoscopic functional lumen imaging probe.

## Data Availability

The datasets used and/or analyzed during the current study are available from the corresponding author upon reasonable request.

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
