# Peer review of "A New Method to Evaluate Lower Esophageal Distension Capacity in Eosinophilic Esophagitis by Using Functional Lumen Imaging Probe (EndoFLIP™)"

_diagnostics, 2024, doi:10.3390/diagnostics14020218_

Round 1
Reviewer 1 Report
Comments and Suggestions for Authors
In this article, the authors present their modified method of Endoflip for EoE evaluation. They stated that the step-wise balloon diameter helps determine the lower esophageal distension capacity of EoE. This is an exciting study in this field of study. However, there are some minor concerns about this article. 1. What kind of endoscope/IEE(e.g., NBI) was used to diagnose EoE? 2. Did experts make the endoscopic diagnosis? 3. The authors could present the Endoflip image and their original step-wise balloon diameter method as Figures to help understand the readers. 4. In Table 1, the P-value is not necessary to describe. 5. There are no explanations of abbreviations at the bottom of the tables.
Comments on the Quality of English Language
Minor English editing is requierd.
Author Response
RESPONSE TO REVIEWER 1
In this article, the authors present their modified method of Endoflip for EoE evaluation. They stated that the step-wise balloon diameter helps determine the lower esophageal distension capacity of EoE. This is an exciting study in this field of study. However, there are some minor concerns about this article.
R: All examinations that led to the diagnosis of EoE in the patients who participated in this study were performed with white light endoscopes. EoE is diagnosed based on the demonstration of a dense, predominantly eosinophilic inflammatory infiltrate in esophageal biopsies, restricted to the mucosa of this organ, in a patient with symptoms of esophageal dysfunction, and once other causes of esophageal eosinophilia have been excluded. (Dellon ES et al. Gastroenterology. 2018;155:1022-1033.e10; Lucendo AJ et al. United European Gastroenterol J. 2017;5:335-358). Certain endoscopic findings are highly suggestive of EoE, but are not considered necessary to establish the diagnosis of the disease.
- Did experts did the endoscopic diagnosis?
R: In general, all patients who provided data for this study were diagnosed and treated for EoE by the physicians who participated as authors in this study.
- The authors could present the Endoflip image and their original step-wise balloon diameter method as Figures to help understand the readers
R: We have followed this reviewer suggestion by including in this new version of the manuscript an explanatory video with a description in the Supplementary material.
- In Table 1, the P-value is not necessary to describe.
R: Thanks for this comment. We have removed the P column in Table 1.
- There are no explanations of abbreviations at the bottom of the tables.
R: The revised version of the manuscript adds this detail.
Reviewer 2 Report
Comments and Suggestions for Authors
This is an interesting study, as it discusses a hot issue. Despite the difficulties that investigators face during the procedure, this method is promising. Apart from the limitations mentioned in this research, using upper digestive endoscopy during the process and examining only the lower esophageal sphincter is a leading work in this situation. More studies with large numbers and multicenter studies are needed to prove these aims.
Comments on the Quality of English LanguageThis is an interesting study, as it discusses a hot issue. Despite the difficulties that investigators face during the procedure, this method is promising. Apart from the limitations mentioned in this research, using upper digestive endoscopy during the process and examining only the lower esophageal sphincter is a leading work in this situation. More studies with large numbers and multicenter studies are needed to prove these aims.
Author Response
RESPONSE TO REVIEWER 2
This is an interesting study as it discusses a hot issue. Despite the difficulties the investigators face during the procedure, this method is promising. Apart of the limitations mentioned in this research, using upper endoscopy during the process and examining only the lower esophageal sphincter is a leading work in this situation. More studies with large numbers and multicenter studies are needed to prove this aims.
We sincerely appreciate your review of our manuscript and comments. Certainly our method represents the first alternative proposal to the only method so far available to interpret Endoflip logs. We agree that our analysis strategy should be evaluated in larger series of patients, and the reproducibility of our method should be evaluated in other series of patients recruited in different settings.
Reviewer 3 Report
Comments and Suggestions for Authors
The article of Dr. Casabona-Frances et al. present a novel technique of functional evaluation of the esophagus by using a modified approach of Endoflip, in the intend to overcome some of the shortenings of the classical approach. The authors describe a new method to evaluate esophageal distension capacity 88 through the analysis of EndoflipTM recordings which makes unnecessary the use of the 89 standard and de facto established protocol of stepwise increasing volumes. They comparatively analyze the results on 2 groups of patients, the first - a control group and the second diagnosed with active eosinophilic esophagitis. The technique and patients' selection criteria are clearly explained and replicable. The statistical analysis presents comparatively the 2 groups in terms of clinical, endoscopical and histological data. The comparative endoscopic and histological findings are then presented for the EoE group, before and after treatment with budesonide orodispensable tablets.
The main drawback of the study is the limited amount of patients included, 14 in each group. As the authors correctly acknowledge, the technique need further studies for validation.
AS minor issues, Discussions could be improved by rephrasing lines 411-415, so no bullets would be used in the text. More studies on the topic could be critically discussed, updating the references accordingly.
Author Response
RESPONSE TO REVIEWER 3
The article of Dr. Casabona-Frances et al. presents a novel technique of functional evaluation of the esophagus by using a modified approach of Endoflip, in the intend to overcome some of the shortenings of the classical approach. The authors describe a new method to evaluate esophageal distension capacity 88 through the analysis of EndoflipTM recordings which makes unnecessary the use of the 89 standard and de facto established protocol of step-wise increasing volumes. They comparatively analyze the results on two groups of patients, the first – a control group and the second diagnosed with active eosinophilic esophagitis. The technique and patients’ selection are clearly explained and replicable. The statistical analysis presents comparatively the 2 groups in terms of clinical, endoscopical and histological data. The comparative endoscopic and histological findings are then presented for the EoE group, before and after treatment with budesonide orodispensable tablets.
We sincerely appreciate the detailed description of this reviewer of our manuscript.
The main drawback of the study is the limited amount of patients included, 14 in each group. As the authors correctly acknowledge, the technique needs further studies for validation.
We agree. These results must be validated in a larger series of patients, and our group is already working on it.
AS minor issues, Discussion could be improved by rephrasing lines 411-415, so no bullets would be used in the text. More studies in the topic could be critically discussed, updating the references accordingly.
Thanks for the suggestion. We have rephrased these Discussion paragraphs to eliminate bullet points as suggested.
To the best of our knowledge, the unique method developed so far for EndoFLIP analysis is the one by the Chicago group (D.A. Carlson, J.E. Pandolfino, etc.) and the software they have also implemented (http://wklytics.com/nmgi), which is also used almost exclusively by members or collaborators of this group (see for instance reference [20]). This method is cited in our work [12,13] and in fact is the motivation behind our own work and thus, we were unable to find and cite other works of EndoFLOP analysis unrelated with the above mentioned.